# Microbiome-Mediated Strategies to Manage Major Soil-Borne Diseases of Tomato

**DOI:** 10.3390/plants13030364

**Published:** 2024-01-25

**Authors:** Shweta Meshram, Tika B. Adhikari

**Affiliations:** 1Department of Plant Pathology, Lovely Professional University, Phagwara 144402, India; shweta.26662@lpu.co.in; 2Department of Entomology and Plant Pathology, North Carolina State University, Raleigh, NC 27695, USA

**Keywords:** tomato, *Solanum lycopersicon* L., plant–microbiome interactions, root exudates, plant immunity, integrated disease management, *Fusarium oxysporum* f. sp. *lycopersici*, *Ralstonia solanacearum*, *Verticillium dahliae*, *Meloidogyne* spp.

## Abstract

The tomato (*Solanum lycopersicum* L.) is consumed globally as a fresh vegetable due to its high nutritional value and antioxidant properties. However, soil-borne diseases can severely limit tomato production. These diseases, such as bacterial wilt (BW), Fusarium wilt (FW), Verticillium wilt (VW), and root-knot nematodes (RKN), can significantly reduce the yield and quality of tomatoes. Using agrochemicals to combat these diseases can lead to chemical residues, pesticide resistance, and environmental pollution. Unfortunately, resistant varieties are not yet available. Therefore, we must find alternative strategies to protect tomatoes from these soil-borne diseases. One of the most promising solutions is harnessing microbial communities that can suppress disease and promote plant growth and immunity. Recent omics technologies and next-generation sequencing advances can help us develop microbiome-based strategies to mitigate tomato soil-borne diseases. This review emphasizes the importance of interdisciplinary approaches to understanding the utilization of beneficial microbiomes to mitigate soil-borne diseases and improve crop productivity.

## 1. Introduction

The tomato is an essential vegetable, and 186 million tons are produced worldwide [1]. The top four tomato producers are China (mainland) with 68.2 million tons, India with 20.7 million tons, Turkey with 13 million tons, and the USA with 10.2 million tons, according to the Food and Agriculture Organization (FAO) data (https://www.fao.org/faostat/en/#data/QCL, accessed on 14 January 2024) for the 2022 year [1,2]. In the United States, tomatoes are grown all year round in California, except for the winter season when greenhouse cultivation is commonly practiced [3]. The tomato growing season in Florida is from October to June; in other states, tomato production occurs during summer [4].

The tomato is susceptible to over 200 diseases caused by various pathogens and pests [5]. Among them, soil-borne diseases seriously threaten the sustainability of tomatoes and food security [6]. Historically, several management strategies have been employed to manage tomato soil-borne diseases. These include the use of resistant cultivars, chemicals (e.g., fungicides and soil fumigants), physical methods such as soil solarisation and soil heating, cultural methods (e.g., crop rotation field sanitation), and biological control [7,8,9]. Among these methods, biological control has recently gained more attention due to its high efficiency, target specificity, sustainability, and public acceptance [5,10,11].

The term ‘microbiome’ is coined by combining “micro” and “biome” to indicate a distinct microbial community present in a well-defined habitat with specific physio-chemical properties or a microbial community associated with a specific plant species or specific crop genotypes [12,13]. Biological control, also known as biocontrol agents (BCAs), refers to the use of living microorganisms or beneficial metabolites synthesized by microbes or microbial gene products to combat specific plant diseases or pests that affect plants, and it involves deploying mechanisms such as antibiosis, parasitism, or nutrition for resources or space [14]. ‘Microbiome-mediated strategies’ involve using the root and soil microbiome to reduce soil-borne diseases and improve plant fitness, immunity, and productivity [5,15]. Plants are intimately associated with many microbiomes that colonize all their root/surrounding roots, collectively known as the plant microbiota [16]. Root microbiota is mostly derived from the soil environment, which contains highly diverse microbiomes dominated by acid bacteria and fungi [17]. Notably, microbiomes significantly enhance plant health by facilitating nutrient accessibility, promoting growth under stresses, providing a natural defense against diseases and pests, and improving soil quality [18,19]. This symbiotic relationship between plants and microbes holds promise for sustainable agriculture, potentially reducing chemical inputs while elevating productivity [20].

Root microbiomes are essential in plant health and disease resistance [21,22,23,24]. Several studies have reported similar findings in rice [25], maize [26], potatoes [27], and tomatoes [28,29,30]. The soil surrounding plant roots, known as the rhizosphere, is filled with billions of microorganisms in just a gram, including tens of thousands of different species [13]. This complex ecosystem constantly changes and is the foundation for exploring microbiome-based strategies to improve crop health [18,31,32], including tomatoes [33].

Microbiome-mediated strategies encompass various approaches. For instance, ‘microbial inoculants’ constitute a main strategy that inoculates specific beneficial microbiomes to the root zone to boost growth, nutrient, and stress tolerance in tomato plants [34]. Another tactic is engineering microbial consortia, which support plant health and provide disease resistance to root pathogens [35,36]. ‘Biofertilizers’ refer to the use of beneficial bacteria in tomato seeds, which promote plant growth [37]. Additionally, the BCA strategy utilizes beneficial microbiomes, such as *Pythium oligandrum* [38], *Trichoderma*, and *Hypocrea* isolates [39], to manage soil-borne pathogens. A strategy to enhance plant immunity has been demonstrated in tomatoes through microbiome transplants [12,40]. This review discusses the potential use of microbiomes to improve the soil health, foster plant growth and disease resistance, and promote the sustainability and resilience of tomatoes while addressing challenges and opportunities.

## 2. Major Soil-Borne Diseases of Tomato

Soil-borne diseases pose a critical threat to tomato production systems due to their persistent nature and impact on root systems [41]. Among these, Fusarium wilt (FW), Verticillium wilt (VW), and root-knot nematode (RKN) pathogens can survive in the soil for extended periods, hindering an effective crop rotation [42]. Consequently, the diseases compromise root function, impeding water and nutrient absorption, leading to stunted growth and a reduced yield (Figure 1).

The importance of soil-borne diseases in agriculture cannot be overstated due to limited control measures, challenges in early detection, and the adaptability of these pathogens [43]. Managing soil-borne diseases in tomato production using agrochemicals is a significant challenge [42]. The application of pesticides has resulted in disturbances to the ecosystem, threats to human health, harm to aquatic ecosystems, diminished beneficial soil microbiomes, and even the depletion of the ozone layer [44,45].

BW caused by *R. solanacearum* race 3 biovar 2 and FW caused by *F. oxysporum* f. sp. *lycopersici* (Fol) are two of the most severe threats to the tomato industry in recent years [46,47]. Notably, three pathogenic races of these pathogens have been identified based on tomato cultivars possessing significant genes for resistance. Emerging threats to tomato production include *V. dahliae* race 3 [48,49] and GRKN, incited by the invasive pest *M. enterolobii* [50]. These diseases primarily target tomato plants’ vascular system, resulting in the plants’ plugging of xylem vessels, wilting, stunting, root galling, and even death (Table 1). Developing effective and sustainable management strategies to protect tomatoes from these diseases is vital to safeguard food security and sustainability [46].

## 3. Microbiome-Mediated Strategies to Mitigate Major Soil-Borne Diseases of Tomatoes

Pre-pant fumigant methyl bromide has been used to manage tomato soil-borne diseases of specialty crops [51,52]. However, this soil fumigant was banned due to its negative impact on human health, environmental pollution, and the depletion of the ozone layer [53]. Although breeding tomatoes for disease resistance is economical, developing tomato varieties resistant to soil-borne diseases is challenging because polygenes or quantitative trait loci control resistance [48,49,50]. Alternatively, grafting a scion (shoot) to a rootstock can protect plants against pathogens and improve yield [54,55]. Two interspecific tomato rootstocks, ‘Maxifort’ and ‘Beaufort’, have profuse root systems and resist soil-borne diseases in greenhouses and fields [55,56]. These rootstocks are also crucial in determining the bacterial community associated with tomato plants [57]. However, the high production costs of grafting make it challenging to implement [55,56]. Anaerobic soil disinfestation (ASD) with organic amendments could be an alternative to chemical fumigation [58], in which organic amendments are added to the topsoil, saturated with irrigation water, and covered by an impermeable plastic film for 3 to 6 weeks [59,60,61]. ASD with different organic amendments enhances plant growth, microbial diversity, and disease control, increasing the yield [59,60,61,62]. However, the high cost of impermeable plastic films is a significant obstacle to adopting ASD widely [63].

One potential solution is harnessing the microbiomes to reduce or suppress disease severities [64] and enhance the plant immune system [65]. Several studies have shown that beneficial microbiomes can improve disease resistance and nutrient uptake [12,13,66,67]. To effectively protect tomatoes from soil-borne pathogens, it is crucial to understand the mechanisms of microbial communities and develop eco-friendly solutions. (Table 2).

To combat soil-borne diseases in tomatoes, microbiomes consisting of bacteria such as *Bacillus*, *Paenibacillus*, and *Pseudomonas* as well as *actinomycetes*, such as *Streptomyces*, and non-pathogenic fungi, such as *Trichoderma*, *Fusarium*, *Gliocladium*, *Aspergillus*, and *Penicillium*, are used. These microbiomes trigger various mechanisms [47,59]. Recently, crop management practices, such as adding biochar, using ASD with organic amendments, manipulating soil pH, and grafting have optimized the microbiome [55,56,57,58,59,61,62,63,90]. These practices increase the composition, richness, evenness, and diversity of soil and root-associated microbiomes [57,58,90]. Various microbial inoculants, such as biopesticides, bioprotectants, biostimulants, and biofertilizers, can be utilized to suppress pathogens by inhibiting pathogen growth by bioactive substances (antibiosis), competing for iron, modifying microenvironments, interfering with pathogenicity, and involving parasitism [91,92,93]. For example, non-pathogenic fungi from the *Trichoderma* genus, or bacterial spp. of *Bacillus* or *Pseudomonas*, produce various metabolites or signal molecules that can play a significant role in their interaction with plants [30,31]. These compounds stimulate plant growth or antagonize fungal pathogens or nematodes [93,94]. Likewise, mycorrhizal fungi live in soil, help with biogeochemical cycles, and improve soil fertility [95]. The microbiome can modify the plant’s defense mechanism against plant diseases [96,97]. Moreover, plant defense hormones, such as salicylic acid, can regulate the soil microbial communities [98]. Tomato and soil-borne diseases can be a model system to investigate beneficial microbiomes for disease management [61] and ecological sustainability in agriculture [62]. Here, we present some examples of the benefits of microbiomes on major soil-borne tomato diseases.

### 3.1. Bacterial Wilt (BW)

Tomato cv. ‘Hawaii 7996′ is resistant to BW and exhibits the most stable resistance to *R. solanacearum* [18]. Moreover, the microbial community structure of this resistant cultivar was distinct from that of the BW-susceptible cv. ‘Moneymaker’ and harbored a significant proportion of Flavobacteriaceae, Pseudomonadaceae, and Sphingomonadaceae families [18]. Moreover, the assembled genome of Flavobacteria, called the TRM1 strain, was associated with carbohydrate metabolism [18], and specific monosaccharide transporters indicate that TRM1 can reduce the availability of sugars that *R. solanacearum* lectin binds to, thereby reducing disease pressure [18,99]. Increasing the soil pH to 6.45 reduces *R. solanacearum* in tomatoes and increases beneficial microbiomes, including bacteria of *Firmicutes* such as *Bacillus Paenibacillus* and others such as *Flavobacterium* (*Bacteroidetes*) and *Pseudomonas* (*Pseudomonadota*) [100,101]. This suggests that changes in soil pH directly affect the spread of BW disease among tomatoes. A recent review [102] explores microbial and non-microbial strategies for BW management in crop plants.

Certain bacteria can inhibit the virulence of *R. solanacearum* [103]. In particular, *P. aeruginosa*, *P. syringae*, and *P. fluorescens* are some of these beneficial bacteria [104]. They can reduce BW incidence, stimulate tomato plant growth, and suppress the disease by secreting peroxidase and lipoxygenase [103,104,105,106]. *Streptomyces* spp. and other bacteria are effective against bacterial wilt of vegetables [107,108]. Furthermore, potent strains capable of combating *R. solanacearum* have been identified after screening 109 bacterial strains [107,108]. This suggests the importance of microbiomes as biopesticides.

### 3.2. Verticillium Wilt (VW)

Managing VW can be challenging as the current methods are ineffective [60]. However, microbiome-based strategies seem promising. The availability of nutrients in the root exudates determines the ability of bacteria to colonize the rhizosphere [48,60]. A new antimicrobial effector named VdAMP2 has been discovered and utilized to manipulate microbiomes in the soil environment [109]. The pathogenic fungus *V. dahliae* uses effector proteins to regulate the compositions of microbiomes in host plants [109]. The discovery of such molecules may lead to a new source of antibiotic production.

A recent study explored the impact of rootstock genotypes and grafting on bacterial communities in the endosphere and rhizosphere [57]. These bacterial communities were compared to non-grafted and self-grafted plants, and the analysis revealed that rootstock genotypes could indirectly influence bacterial taxa, with certain taxa being associated with specific rootstocks [57]. Additionally, arbuscular mycorrhizal fungi (AMF) and compost can decrease VW severity and boost plant growth, yield, and fruit quality in tomatoes [110].

Endophytic *B. velezensis* strain C2, identified through a genetic analysis, effectively inhibited *V. dahliae* in vitro and in vivo [111]. This strain possesses genes that produce potent antifungal compounds, including lipopeptides (bacillomycin, fengycin, surfactin), and polyketides (e.g., macrolactin, bacillaene, and the dipeptide bacilysin). Other findings suggest that grafting susceptible tomato plants onto commercial rootstocks notably improves their tolerance to VW [112]. The ‘Beaufort’ rootstock significantly reduced VW severity [112]. Another study on grafting susceptible scions onto resistant rootstocks prevented VW symptoms. Gene expression changes were pronounced in susceptible scions, suggesting genes related to pathogen recognition and cell wall modification during tomato–pathogen interactions [113]. Remarkably, strains of *V. klebahnii* (e.g., strain PD659) can potentially be used as BCAs against *V. dahliae* [114].

### 3.3. Fusarium Wilt (FW)

Increasing the diversity of bacterial and fungal communities in the soil around plant roots has been shown to reduce the incidence of FW [115]. To mitigate fungal wilt, a microbiological culturomics approach was employed to identify 205 unique strains and create synthetic communities (SynComs), which were then inoculated into tomato seedlings [115]. A mixture of fungi and bacteria in cross-kingdom SynComs was more effective in suppressing FW than fungi or bacteria alone [115]. Moreover, this effectiveness was attributed to a combination of molecular mechanisms related to plant immunity and microbial interactions, which were contributed by bacterial and fungal communities [115,116]. These findings provide insights into how microbiome dynamics affect plant immunity, and the manipulation of SynComs can be a novel strategy to mitigate FW [116]. Applying a biochar amendment can enhance plant resistance, and the existence of beneficial bacteria such as *Pseudomonas* can protect plants from FW [117]. Moreover, biochar can aid plants in attracting more beneficial bacteria to the rhizosphere, which can facilitate the growth of a disease-resistant microbiome [117].

The use of microbial pesticides is a promising approach to controlling FW biologically. For instance, a BCA can effectively stimulate the activity of defense-related enzymes [118]. Interestingly, *Streptomyces* spp. strain NSP3 triggers the defense mechanisms of tomatoes against *F. oxysporum* f. sp. *lycopersici* [119]. In greenhouse settings, introducing *S. griseus* into the root system before FW infection has proven more effective in controlling FW than tomato seed treatment [120].

### 3.4. Root-Knot Nematode (RKN)

*Meloidogyne* spp. are obligate sedentary plant parasites [121]. The impact of RKN on the host plants can be exaggerated through soil-borne fungi such as *F. oxysporum* and *V. dahliae* [121]. *M. enterolobii* is a significant worldwide threat to food security [50]. Some of the commonly used parasitic fungi include *Aspergillus niger* Tiegh., *Purpureocillium lilacinum* (Thom), *Trichoderma harzianum* Rifai, *T. viride* Pers., and *Metacordyceps chlamydosporia* (H.C. Evans), which have been used as bionematicides against RKN [122]. Additionally, these fungi produce hydrolytic enzymes, such as chitinases, and are responsible for chitin degradation, a structural component of RKN [123].

Tomato plants were treated with a mixture of beneficial BCAs as soil drenches before being infected with *M. incognita* [124]. It was found that the pre-treatment with BCAs caused the plants to be less sensitive to the infection. This BCA-mediated immunity relies on salicylic acid-mediated systemic acquired resistance (SAR) [124]. Importantly it is associated with activating pathogenesis-related genes (PR genes) such as chitinase and glucanase and inhibiting the plant’s antioxidant enzyme system [124].

Other cultural practices can effectively control RKN in soil and roots and improve plant health [125]. Combining soil solarization with resistant tomato cultivars can be a long-term strategy to manage RKN [126]. Another effective method is to use soilless media, such as cocopeat and vermicompost, with organic BCAs. This method has been shown to enhance tomato growth in nurseries and improve plant health, especially in RKN fields [126]. Adding ASD with carbon sources, such as wheat bran, molasses, or ethanol, significantly reduced RKN galling in tomatoes in Ohio, USA, and it increased certain bacterial communities compared to non-amended controls [90].

## 4. Possible Direct and Indirect Beneficial Effects of Microbiome on Tomatoes

The rhizosphere is a dynamic environment with thriving microbial communities [13]. For instance, tomato plants have various beneficial microbiomes, including actinomycetes, bacteria, and fungi, that can significantly impact their growth and functionality [10]. Although there have been some studies on the potential of endophytes to improve plant growth and mitigate plant diseases, there is a lack of comprehensive research on the specific endophytic components and their functions [127]. This knowledge gap hinders our understanding of how endophytic components can benefit plant growth and protect soil-borne pathogens in organic and conventional tomato production systems. To overcome this problem, SynCom is a promising approach to gaining insights into the interactions between plants and their microbiomes [128]. Available evidence suggests that the structure of plant microbiomes is also influenced by the interactions between the plant, microbes, and their physical and chemical environment (Figure 2, [94]).

Microbiomes can enhance plant growth through direct and indirect effects [129,130]. Direct effects can occur when the microbiome produces hormones such as auxin (IAA), cytokinin, and gibberellin or solubilizes nutrients such as phosphorus, iron, or potassium, which in turn results in increased plant biomass, increased yield, and the improved nutritional value of fruits and seeds [5,131,132]. The indirect effects of the microbiome help reduce the negative impact of soil-borne pathogens [129]. This is achieved through ACC deaminase, which lowers ethylene levels, the production of pathogen-inhibiting antibiotics and bacteriocins, the synthesis of pathogen fungal cell wall-degrading enzymes and plant defense genes (e.g., chitinases, glucanases), pathogens that out-compete for nutrients (e.g., siderophore production) or space, the synthesis of fungal pathogen-inhibiting hydrogen cyanide, and the induction of systemic resistance mechanisms such ISR and SAR [129,130,131,132].

## 5. Plant–Microbe Communication

The communication between plants and microbes occurs through molecular signals, chemical compounds, and genetic exchanges, resulting in mutual benefits, such as improved plant health, growth, and resilience [95]. Recent reviews have comprehensively explored how soil and plants affect the composition of rhizosphere microbial communities [97]. The soil type plays a more significant role in determining the structure of microbial communities than the plant type [96,98]. Conversely, plant microbes improve the soil structure and quality by enhancing nutrients, siderophores, and metabolites (antibiotics, toxins, and lytic enzymes) [18,99,133]. Effective communication among microbes and their response to environmental signals is crucial for microbial communities’ success, competitiveness, and dominance (Table 3).

## 6. Hormones and Metabolites Linked to Microbiome Functions

Plant roots release phenolics, amino acids, nucleotides, sugars, terpenoids, polysaccharides, and proteins [138]. Although the composition and abundance of these compounds vary with the plant species, their growth stage, and the presence of stress factors, these metabolites contribute to nutrient availability by promoting microbial growth and solubilizing minerals through root exudates [138]. Furthermore, these metabolites are vital in plant defense mechanisms that help inhibit pathogens and promote resistance [139]. Plant roots also secrete secondary metabolites (SMs) into the soil, affecting the proliferation or suppression of specific microbes around the host plant’s root [140]. Recent studies have shown that selected SMs, including coumarin, triterpenes, flavonoids, benzoxazinoids, and phytohormones, can shape the composition and functions of the plant’s microbiome [139]. The exudation of metabolites is also linked to microbiome functions and can shape microbiome composition [115,116,139]. Some prominent metabolites include glucosinolates, flavonoids, coumarins, benzoxazinoids, triterpenes, and defense-related phytohormones jasmonate and salicylate [112]. Plants can adapt to pathogen infection by changing the chemistry of their root exudation [117], which may help them to assemble microbiomes that promote plant health [12].

Plants form a mutually beneficial relationship with microbiomes in their rhizosphere to deal with biotic stresses [141]. Volatile organic compounds (VOCs) play an important role in this process, and the recruitment of beneficial microbiomes induced by biotic stress can further strengthen the plant’s defense mechanism [141]. Plants can communicate with different microbiomes using VOCs and play a crucial role in ISR in plants [142]. Various strains of PGPR can produce different types of VOCs, which can trigger a systemic defense against pathogen attacks in plants [143]. It has been found that even plants separated by distance can communicate with each other by releasing certain airborne VOCs. The rhizosphere microbiome of a tomato plant treated with a PGPR, *B. amyloliquefaciens* GB03, on one plant could affect the rhizosphere microbiome of a neighboring plant through microbe-induced plant volatiles (MIPV) [144]. They found a high degree of similarity (up to 69%) in the inoculated and neighboring plants’ microbial communities of the rhizosphere [144]. When the leaves of a tomato plant were treated with strain GB03, it released β-caryophyllene, which served as a signature VOC [144]. This resulted in the release of a large amount of salicylic acid (SA) in the root exudates of a neighboring tomato seedling. This is the first demonstration that an MIPV derived from PGPR has been found to modulate the rhizosphere microbiome of spatially distant plants [144].

Plant roots can recruit microbiomes that protect them from different stresses and improve their growth [145,146]. Disease-suppressive soil activity can also be developed in high-intensity production systems that rely on successive crop monocultures [147]. A well-known example is the take-all decline when wheat is grown continuously in soil infested with the pathogenic fungus *Gaeumannomyces graminis* pv. *tritici* [147]. Initially, this leads to an increase in the disease, but over time, the disease gradually declines. Host plants can actively recruit or stimulate biocontrol agents to resist pathogen invasion, and this is known as the ‘cry for help’ hypothesis [145,146]. Studies have confirmed that enriching soil microbiomes can suppress diseases [146,148]. Plants vary in physiology and immune responses to pathogen invasions [141]. Furthermore, the molecular mechanisms and chemical basis of the ‘cry for help’ strategy may vary depending on the plant species [122]. Disease-resistant plant genotypes also accumulate beneficial rhizosphere microbiomes, indicating a connection between plant immunity and microbiome composition [149].

The complex interplay involves the subversion of plant hormone signaling and immune responses by specific microbial effectors [150,151]. Plant-derived vital hormones are released into the rhizosphere and can directly affect plant-interacting microbiomes and the root-associated microbiome (Table 4).

## 7. Enhancing Plant Immunity through Microbiome Engineering

To investigate the impact of organic amendments on bacterial communities in the rhizosphere and soil-borne pathogens, researchers performed 16S rDNA amplicon sequencing analysis [168]. They found that vermicompost influences the microbiome assembly of the rhizosphere by directly colonizing vermicompost-origin taxa and inducing changes in the rhizosphere microbial community [168]. For instance, Kwak et al. [18] analyzed the rhizosphere microbiomes of tomato-resistant cv. ‘Hawaii 7996’ and susceptible cv. ‘Moneymaker’ to determine whether plant-associated microbiomes can contribute to disease resistance to *R. solanacearum*. Their findings suggested that the flavobacterium strain TRM1 can suppress *R. solanacearum* in a susceptible plant [18]. Furthermore, they discovered that certain characteristics of strain TRM1, related to carbohydrate metabolism and other functions, could help it adapt to life in the tomato rhizosphere while protecting it against *R. solanacearum* [18]. Interestingly, specific monosaccharide transporters suggest that TRM1 could reduce the availability of sugars that *R. solanacearum* lectin binds to, thus lowering the risk of infection [16,18,70,91,169,170].

In a study, researchers tested the hypothesis that microbiomes are associated with resistance to *R. solanacearum* in tomato recombinant inbred lines (RILs) [171]. They found that certain bacterial species present in the microbiome of different tomato genotypes can enhance plant growth. However, the selection of rhizosphere and endosphere bacterial species is influenced by the specific tomato genotype. Different tomato genotypes can exhibit varying degrees of defense against *R. solanacearum* based on their microbiome composition [171]. Another study provided evidence that bacterial communities associated with plant roots, particularly those containing eight *Pseudomonas* spp. strains producing the antibiotic 2, 4-diacetyl phloroglucinol (DAPG), play a crucial role in suppressing *R. solanacearum* in the tomato rhizosphere [172]. Furthermore, high Pseudomonas diversity reduces pathogen density, thus decreasing disease incidence through intensified resource competition and interference with the pathogen [172].

A recent study on *Arabidopsis* showed significant genomic differences in cultivated *Pseudomonas* spp. [173]. The study identified the genes responsible for evading plant defenses and inducing susceptibility, which suggested the molecular mechanisms underlying plant–microbe interactions [119,173,174]. Adding biochar significantly improved the disease-suppressive properties of the microbiome against FW [175] and caused significant alterations to the root microbiome [176]. The ultimate goal is to improve tomato–microbe interactions by integrating microbial biofertilizers, BCAs, soil amendments, grafting, optimized microbiomes, and matching microbe-optimized tomatoes for different soil types (Figure 3).

However, this area requires significant research efforts to enhance plant immunity and yields, boost food security, and promote eco-friendly practices [12,115,127]. The current microbial technologies and emerging microbiome-related approaches offer new, more sustainable ways to increase crop productivity [12]. A way to improve plant health is to select and evolve microbiomes that have specific effects on host fitness using a top-down strategy [177]. Host-mediated microbiome engineering indirectly selects microbial communities through the host and uses host traits that affect microbiomes [178]. Studies have demonstrated that phytohormones can be used to optimize microbiomes for disease resistance, and these optimized microbiomes can be utilized as inoculants to provide disease resistance or plant immunity [177,178]. Plant diseases can be mitigated by creating a host-mediated selection of microbiomes without genetically modified microbiomes [178]. In addition, the concept of the “functional core microbiome” and the development of SynComs have emerged as the genome engineering toolkit to manipulate microbial communities with high specificity and efficacy over various specificities and magnitudes [12,179].

## 8. Omics and Novel Technologies to Investigate Tomato Microbiome

### 8.1. Omics Technologies

Integrating multi-omics and high-throughput sequencing technologies can unlock discoveries of microbial ecology, plant–microbiome interactions, and microbiome–microbiome interactions [180]. Currently, the most commonly used omics approaches for identifying and characterizing microbiomes’ diversity and functionality are *16S ribosomal* RNA (16S rRNA) and internal transcribed space (ITS) gene sequencing, metagenomics, metabolomics, metatranscriptomics, and metaproteomics (Table 5).

### 8.2. Harnessing Microbiome through Genome Editing

The native plant microbiome has enormous potential for selecting advantageous microbial functions [184]. Different crops have distinct microbiome members that consistently exist within a core group across various environments [184]. These members interact mutually and beneficially with the broader plant-associated microbiomes [185]. Maize, rice, and sugarcane are crops that have successfully harnessed their core microbial communities [186]. Similarly, science-based evidence suggests that tomatoes have a genotype-dependent core microbiome, indicating that this approach can enhance tomato root microbiomes.

Clustered, regularly interspaced short palindromic repeat (CRISPR) technology is valuable in studying microbiomes [187]. Among the different types of CRISPR systems, type II was beneficial for targeted genome editing in prokaryotes [188]. With the use of integrative plasmid in CRISPR-Cas9 edited *B. subtilis*, horizontal gene transfers are significantly reduced, thus minimizing the risk of horizontal transfer [186]. For instance, two rhizosphere-associated bacteria, *B.subtilis* HS3 and *B. mycoides* EC18, have been found to promote plant growth [189]. Additionally, the CRISPR-Cas9 system was utilized to investigate the plant–microbe interaction mechanisms. They found that *B. subtilis* HS3 utilizes fengycin and surfactin to exhibit antifungal activity, and this strain also releases other volatile organic compound 2, 3-butanediol, thus promoting plant growth [189]. Plants and microbiomes use signaling molecules to communicate their needs to each other [190]. For example, when plants are low in phosphorus, they release signal particles that cause rhizosphere organisms to upregulate their phosphorus-assembling genes. These signaling molecules or genes are also released by plants when pathogens attack them [12]. Overall, identifying signaling genes and combing them through genome editing is essential for their practical use in sustainable agriculture.

### 8.3. Biochemical Approach

Plants release chemical signals and substances from their roots to enhance the root microbiome, promoting a more beneficial microbiome makeup [181]. These exudates comprise organic acids, amino acids, sugars, alcohols, phenolics, polypeptides, enzymes, proteins, and hormones [181]. Root exudates shape plant–environment interactions [191]. Analyzing these exudates through advanced metabolomics is advantageous for investigating the interplay between plant–soil-borne pathogen interactions and optimizing plant–soil dynamics [192,193,194].

Plants can selectively promote the growth of beneficial microbiomes by changing the profile of metabolites released from the roots. This method enhances nutrient absorption, disease resistance, and overall plant health by promoting a more beneficial microbiome makeup [195]. The soil microbiome, through nutrient cycling [196] and pH regulation [41], improves soil structure significantly, which influences soil chemistry and mitigates soil-borne diseases in tomatoes [15]. Microbial activities, including decomposition, mineralization, and chemical signaling, enhance plant health and contribute to a balanced and beneficial rhizosphere [191].

Plant growth-promoting microbiomes (PGPM) influence the soil’s oxidative, pH, and ion levels, shaping the chemical environment conducive to plant growth and development [197]. These findings lay the groundwork for exploring microbiome-based strategies to optimize soil conditions and combat soil-borne pathogens. For instance, studies reveal that reducing soil pH compromises the natural defense mechanisms of the soil microbiome against pathogenic Fusarium infections [198].

### 8.4. Synthetic Microbial Community (SynCom)

SynComs, which are tools for rhizosphere engineering, have an advantage over conventional bioinoculants as they are more robust and similar to the natural interactions in the rhizosphere [199]. Additionally, we must explore more strategies for designing and optimizing SynComs, which may lead to more reliable assemblages for combating biotic stresses in agriculture and improving performance. To achieve the desired SynComs function, microbial biofilm formation, production of secondary metabolites, ability to induce plant resistance, and the dynamics between plants and microbiomes need to be investigated [200]. This involves cultivating microbial consortia to explore how plant–microbe interactions impact growth and development [201]. Using these methods, researchers can unravel the regulatory mechanisms plants employ to shape their microbiomes and vice versa [202].

To develop SynComs, one suggestion is to culture microbiome fractions (MF) from cv. ‘Hawaii 7996’ rhizosphere in a modified M9 medium, mimicking tomato root exudates [203]. Successive transfers led to upland MM9 1st transfer, showing bacterial wilt suppression. Dominant Proteobacteria (98.12%), primarily Enterobacteriaceae (86.76%) with Enterobacter genus, were identified. This approach yields potential biocontrol agents, highlighting the MM9 medium’s success in cultivating disease-suppressive rhizosphere microbiota [203]. Recent studies discovered that combining *B. subtilis* J3 and *P. fluorescens* J8 in a synthetic community effectively controlled *Alternaria solani* in tomatoes [204,205]. BCA inhibited the growth of *A. solani* by 94.91%, reduced disease, promoted growth, and modified the microbial community, thus increasing the number of beneficial bacteria [205]. Moreover, a study on 205 strains of SynComs found that higher diversity in rhizosphere bacterial and fungal communities was associated with a lower incidence of FW in tomato seedlings [115]. Additionally, cross-kingdom SynComs, which combined plant immunity and microbial interactions, were the most effective in suppressing the disease [200].

### 8.5. DefenseBiome

Liu et al. [31] introduced a new concept called ‘DefenseBiome,’ which encourages the design of SynComs by combining metagenomics and metabolomics analyses. The main goal is to create a culture bank of stress-induced microbiomes and incorporate their constitution into SynComs. For example, strains of *Xanthomonas*, *Stenotrophomonas*, and *Microbacterium*, isolated from the rhizosphere of *Arabidopsis thaliana*, interacted synergistically to form biofilms, which helped develop ISR against *Hyaloperonospora arabidopsidis* [201]. When left to grow independently, these bacteria did not significantly affect the plant. However, if there was a previous population of plants infected with downy mildew in the same soil, it provided better protection against this pathogen for a second population of plants [201]. This indicates that plants manage their rhizosphere microbiome to draw in a specific microbiome that can trigger plant defense mechanisms [206].

### 8.6. Culturomics

Microbiomes are mainly studied through metagenomics, which can only detect bacteria that cannot be cultured [207]. To overcome this limitation, researchers have developed a new method called culturomics, which involves designing specific culture media that cater to particular microorganisms and applying selective and/or enrichment culture conditions [202]. This approach can help isolate fastidious human gut microbiomes and detect minority populations. MALDITOF-MS and 16S rDNA identification identify the bacterial strains [207]. Culturomics can help describe new species and study the interaction between different bacterial strains within a given microbiota [202]. This approach has also been applied to plant microbiomes, where culture collections can help identify bacterial genes associated with plant nutrition, pathogenicity, and colonization [208].

Although culturomics has emerged as a novel approach combining next-generation sequencing and using a wide range of culture media [202,208], it has not yet been extensively used to describe the tomato microbiome. One study used tomato juice to cultivate the bacterium *Oenococcus oeni* [209]. In another study, various vegetables, including tomatoes, were sampled and analyzed, and the rhizosphere soils of various vegetables, including tomatoes, were sampled and analyzed to understand microbial diversity [210]. High-throughput culture platforms that use microdroplets and microfluidic technology can improve isolation efficiency by allowing extensive microbial isolate separation, genome sequencing, and functional screening [207,211].

## 9. Prospects for the Use of Microbiome-Mediated Strategies

Microbiome-based strategies have shown promise in managing soil-borne pathogens in tomato agri-biotechnology. These approaches aim to improve disease resistance and nutrient uptake while reducing chemical dependence. However, some challenges must be addressed before their broader implementation. Technological barriers hinder the manipulation of entire microbial communities, and there is limited understanding of microbial interactions. To overcome these challenges, it is important to integrate microbiome-mediated approaches into agriculture. Future research should prioritize unraveling plant–microbe interactions using advanced metagenomic, metabolomic, metatranscriptomic, and proteomic approaches and employing high-throughput sequencing technologies. Exploring microbial strain specificity and its effects on soil-borne pathogens across diverse environments will create robust agri-biotechnological solutions. However, we still need to discover the untapped potential of soil microbiomes, their biological functions, their stability, their scalability, and how they work together as a community.

We can use natural or engineered soil inoculants and consortia to improve crop performance and mitigate the negative consequences of disease epidemics. Manipulating root exudates and utilizing SynComs, which is the inherent potential of the native plant microbiome, can also be helpful. Combining microbiome engineering and plants with the best management practices is necessary to address climate change and growing populations, promote ecosystem sustainability, and improve food security.

## Figures and Tables

**Figure 1 plants-13-00364-f001:**
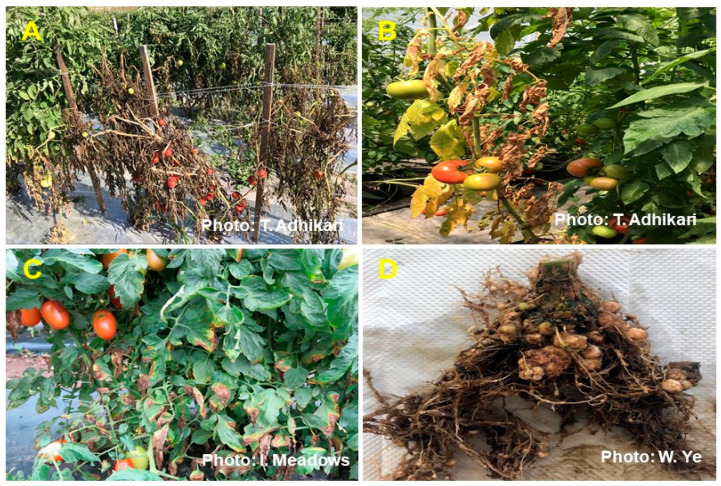
The major soil-borne diseases affecting tomatoes. These pathogens infect susceptible plant tissues or can enter through the roots, causing typical symptoms such as wilting, stunting, yellowing, and root galling. These symptoms can lead to premature defoliation and yield reduction. The diseases include BW caused by *Ralstonia solanacearum* (**A**), FW caused by *Fusarium oxysporum* f. sp. *lycopersici* (**B**), VW caused by *Verticillium dahliae* (**C**), and GRKN caused by *Meloidogyne enterolobii* (**D**).

**Figure 2 plants-13-00364-f002:**
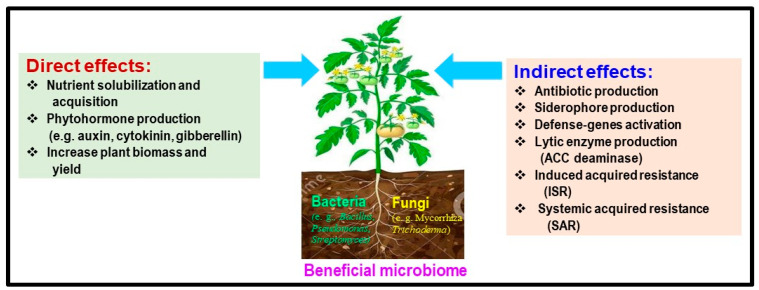
Possible direct and indirect mechanisms of the microbiome can enhance plant growth, increase yield, and control soil-borne diseases in tomatoes. These effects are further detailed in Table 2.

**Figure 3 plants-13-00364-f003:**
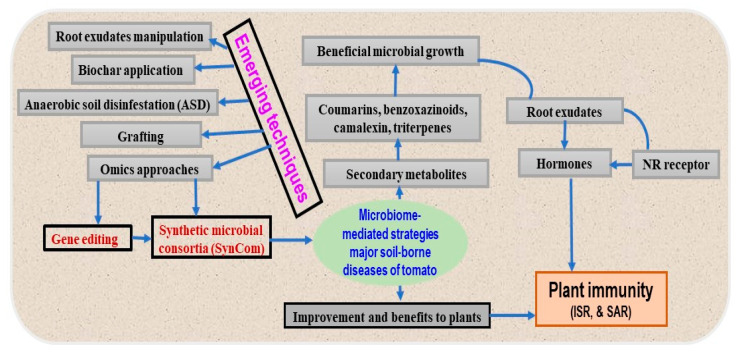
Possible integrated approaches to investigate the beneficial effects of microbiome and enhance plant immunity against soil-borne pathogens of tomatoes. The microbiome can benefit any part of a plant and transmit this benefit to other parts via plant-mediated transport or a signal. Multi-omics approaches, next-generation sequencing technologies, and microbiome engineering provide new opportunities to explore these benefits.

**Table 1 plants-13-00364-t001:** Major soil-borne diseases of tomato production.

Diseases	Causal Agent	Consequences	References
Bacterial wilt (BW)	*Ralstonia solanacearum*	Seed and soil-borne disease. Severe under high-temperature. Humidity and low soil pH.	[46]
Fusarium wilt (FW)	*Fusarium oxysporum* f. sp. *lycopersici*	Cause epinasty and drooping and dark brown.	[47]
Verticillium wilt (VW)	*Verticillium dahliae*	The xylem vessels are blocked, preventing water and nutrient uptake preventing water and nutrients uptake, causing impairment in the plant’s vital functions.	[48,49]
Root-knot nematode (RKN)	*Meloidogyne* spp.	Associated with galling and rhizosphere damage, soil-borne pest. Root galling and defeats most *Mi*-genes for resistance	[50]
Guava root-knot nematode (GRKN)	*Meloidogyne enterolobii*		[50]

**Table 2 plants-13-00364-t002:** Some mechanisms of microbiome-mediated strategies for managing major soil-borne tomato diseases.

Mechanism	Disease/Pathogen/Strategies	References
Bacterial antagonisms	*B. subtilis* QST713 has effectively controlled BW by producing antifungal metabolites.	[68]
Induced systemic resistance (ISR)	WF02 showed efficacy against BW in diverse tomato cultivars by activating common defense genes (phenylalanine ammonia-lyase and PR1a while specifically inducing the jasmonic acid pathway in susceptible cultivars.	[69]
Microbiome manipulation	Incorporating bio fertilizer *S. microflavus* G33 is also an effective biocontrol agent for mitigating tomato bacterial diseases. Applying compost or vermicompost increases beneficial microorganism abundance, reducing BW in tomato crops.	[70,71]
Phage therapy	The bacteriophage ɸsp1 exhibits the potential for creating a biocontrol formulation to control *R. solanacearum V. dahliae* and *F. oxysporum* sp. f. *lycopersici.*	[72]
Fungal antagonism	AMF and *T. harzianum* are effective BCAs against FW and VW.	[73]
Bacterial antagonism	*B. amyloliquefaciens* Oj-2.16, *B. subtilis* NCD-2 (BS), *B. cereus* MH778713: Produce antibiotics, enzymes, and other metabolites that can kill FW pathogen. *P. putida* triggers innate resistance against FW in tomato through the activation of PR protein and phenylpropanoid pathway. *P. fluorescens* NZ105 produces siderophores that chelate iron, making it unavailable to the FW pathogen.	[74,75,76,77,78]
ISR	*B. subtilis* MBI600, *B. subtilis* FB17 can trigger ISR in tomato plants. *Pseudomonas* spp. produces VOCs as elicitors of transcription factor gene MYB72 has been reported in ISR.	[79,80,81]
Microbiome manipulation	The addition of biochar to soil enhances microbial biomass, promotes plant growth, and mitigates FW wilt in tomatoes by inhibiting the growth of fungal mycelium and reducing the germination of microconidia.	[82,83]
Rhizome microbiome engineering	Seed priming with nanoparticles, beneficial microbiomes, and PGPR elicited a defense mechanism in response to wilt disease in tomato.	[77,84]
Microbial antagonists	Several beneficial microorganisms, including bacteria, fungi, and actinomycetes, can directly antagonize RKN. *B. subtilis* strain Bs-1, MTCC441, has demonstrated significant nematicidal activity against *M. incognita*. Volatile organic compounds of *B. cereus* strain Bc-cm103 show an antagonistic effect.	[85,86,87]
AMF-mediated antagonism	These fungi can enhance nutrient uptake by tomato plants and suppress RKN by acting as physical barriers or producing antifungal compounds. MF with hydroxyapatite nanoparticles or *Trichoderma* can effectively control RKN.	[88,89]

**Table 3 plants-13-00364-t003:** The types of communication between microbiomes and their effects on tomato plant–microbe interactions.

Type of Communication	Mechanism(s)	Remarks	References
Bacterial cell-to-cell communication	Quorum sensing (QS)	Inhibiting QS reduces the virulence of *R. solanacearum* species complex by controlling EPS and secondary metabolite production, biofilm formation, and cellular motility. PQI-5 and other antagonists notably decreased wilting symptoms in tomato plants infected with strain OE1-1.	[134]
Fungal cell-to-cell communication	Effector-triggered immunity (ETI) during tomato—*F. oxysporum* interaction	*Avr2* and *Six5* from *F. oxysporum* trigger ETI in tomatoes. They interact at plasmodesmata, enabling *Avr2* movement with *Six5*, causing disease in susceptible plants. In resistant plants, I-2 protein recognition of *Avr2* in xylem-adjacent cells induces a hypersensitive response (HR).	[135]
Cross-domain and cross-kingdom communication	Inter-domain communication in symbiotic relationships in tomato during FW	Diverse rhizosphere bacterial and fungal communities were strongly linked to lower FW incidence. Cross-kingdom SynComs (fungi and bacteria together) were more effective in suppressing FW than separate fungal or bacterial SynComs.	[115,136]
Effects of multipartite interactions on plants	Multipartite interactions impact plant growth. Microbe interactions shape root surface microbial communities.	Soil type influences *Pseudomonas* spp. diversity in tomato plants, suggesting soil characteristics dictate dominant rhizosphere microbes.	[137]

**Table 4 plants-13-00364-t004:** The hormones and metabolites that are involved in modifying the tomato root microbiome.

Metabolite	Examples	Impact on Microbiome	References
Coumarins	Scopoletin	Modulate root microbiome composition and exhibit antimicrobial effects on specific fungi.	[152]
Benzoxazinoids	DIMBOA	Attract beneficial bacteria; influence microbiome composition.	[152]
Camalexin	Phytoalexin	Modulates microbial strains; impacts plant–microbe interactions.	[153]
Triterpenes	Diverse types	Modulate microbial growth; affect root microbiota. Candidate genes for novel sesquiterpene biosynthesis have been identified in wild tomatoes.	[154,155]
Aromatic organic acids, glycoalkaloid	Nicotinic, shikimic, and salicylic, tomatin	Influence microbial community assembly and provide substrates for bacterial growth.	[156,157]
Other metabolites	Azelaic acid, hexanoic acid	Transmit systemic changes in exudates and shape the microbial community. They have been suggested to trigger plant defense through a priming mechanism in tomato and enhance broad-spectrum defense.	[153,158]
Indole-derived metabolites	Indolic compounds	Impact microbial functionality and abundance Related to Systemic Acquired Resistance in tomato.	[144,159]
Strigolactones (SLs)	Arbuscular mycorrhizal fungi (AMF), *Rhizobia*	Functions in shoot branching, root architecture. Dual role as host detection signals and germination in tomato.	[160,161]
	*F. solani*, *Rhizobiaceae*, *Bdellovibrio*, *Shinella*	It affects microbial community composition.	
	*Olpidium brassicae*	Complex interactions with other hormones.	
Abscisic acid (ABA)	*Rhodococcus* spp., *Novoshingobium* spp. *Aspergillus nidulans*, *Saccharomyces cerevisiae*	Involved in stress tolerance, seed dormancy.	[162]
	*Streptomycetaceae*	Used as a carbon source by some rhizobacteria in tomato.	[119]
Auxin (IAA)	*Streptomycetaceae*	PGPR induces organic volatile compounds that increase the expression of auxin synthesis genes (SlIAA3 and SlIAA1) and elevate the auxin levels within tomato plants. Used as a carbon source by some rhizobacteria.	[160,163]
Jasmonic acid (JA)	Actinobacteria and beneficial microbial taxa	Defense hormone interacts with the SA pathway rhizosphere of tomato. Affects rhizosphere community diversity.	[138,151]
Salicylic acid (SA)	Beneficial and pathogenic microbial taxa	The defense hormone interacts with the JA pathway. Impacts endosphere and rhizosphere communities.	[164,165]
Ethylene (ET)	*Actinobacteria*	Defense hormone, synergistic with JA and related to immune signaling. Endosphere communities of tomato associated with degradation of the ET precursor 1-aminocyclopropane-1-carboxylic acid (ACC).	[166,167]
Cytokinin (CK)		Involved in nodule formation and pathogen resistance. PGPRs promote the production of CK and other growth regulators in tomato.	[151,165]
Gibberellic acid (GA) and brassinosteroids (BR)		Impacts nodulation, and AMF formation.	[138]

**Table 5 plants-13-00364-t005:** The use of omics approaches to investigate microbiomes.

Techniques	Remarks	References
Amplicon sequencing (16S for bacterial communities and ITS for fungal communities)	Discovers and screens genetic variations used for taxonomy and phylogeny profiling	[180]
Metagenomics	Identifies taxonomic composition and microbial diversity	[180]
Metabolomics	Metabolomics uncovers specific molecules plants release, revealing interactions with root microbiome and ecosystem. This approach can reveal the metabolic profile of root exudates and soil during host–pathogen interactions	[181]
Metatranscriptomics	Provides insights into gene expression patterns of plants and their microbiome. Helps to understand how microbes influence plant health and stress responses. It helps identify critical genes, pathways, and molecular signals in plant–microbe interactions, aiding the discovery of biocontrol agents and growth promoters	[116,182]
Metaproteomics	Metaproteomics analyses microbial proteins in the tomato root microbiome, providing insights into essential microbes, responses to environmental changes, and ways to optimize plant–microbe interactions for better plant health	[183]

## Data Availability

Not applicable.

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
