# Peer review of "Microbiome-Mediated Strategies to Manage Major Soil-Borne Diseases of Tomato"

_plants, 2024, doi:10.3390/plants13030364_

Round 1
Reviewer 1 Report (New Reviewer)
Comments and Suggestions for Authors
The authors of the manuscript entitled: “Microbiome-mediated strategies to manage major soil-borne diseases of tomato” have described recent data concerning this interesting an valuable theme.
In my opinion, the review paper has been prepared with high scientific quality and is very well presented. Thus it can be published in the current version.
I have only one minor suggestion concerning Tables 2 and 4. The lines between some rows are redundant and should be removed. For example, below Phage therapy, Rhizome microbiome engineering, below Auxin, Jasmonic acid etc. Also, missing literature citing should be added into these table references.
Author Response
January 17, 2024
Mr. Lance Cheng
MDPI Plants Editorial Office
St. Alban-Anlage 66, 4052 Basel, Switzerland
Dear Mr. Cheng:
Greetings from NC State University.
We are pleased to resubmit the manuscript "Microbiome-mediated Strategies to Manage Major Soil-borne Diseases of Tomato" with the Manuscript ID: plants-2816729 for publication. We appreciate your efforts and time in reviewing our manuscript and providing valuable feedback that helped us improve it. We have considered the comments from you and the other two reviewers and made several positive changes in this revised version. Some of the main changes include:
- Restructuring the Abstract, and Introduction.
- Polish and clarify the language in all sections to avoid any confusion.
- We added new Fig. 1C and 1D and revised Fig. 2 and Fig. 3 to better reflect the microbiome's roles in tomato diseases and plant immunity.
- Describe the direct and indirect effects of microbiomes in mitigating major soil-borne diseases of tomatoes. Additionally, we have included several new references in the revised manuscript.
We edited the manuscript before submission and attached the following documents.
- A rebuttal letter that responds to each point raised by the two reviewers.
- A "revised manuscript” - changes are highlighted in yellow.
I appreciate your consideration. We look forward to hearing back from you soon.
Best regards,
Sincerely
*****************************************************************************
Responses to Reviewer #1’s Comments:
Comments and Suggestions for Authors:
GENERAL COMMENTS: The authors of the manuscript entitled: “Microbiome-mediated strategies to manage major soil-borne diseases of tomato” have described recent data concerning this interesting an valuable theme.
In my opinion, the review paper has been prepared with high scientific quality and is very well presented. Thus it can be published in the current version.
RESPONSE: We value your insightful feedback and would like to express our gratitude for your excellent assessment. Thank you very much.
COMMENT: I have only one minor suggestion concerning Tables 2 and 4. The lines between some rows are redundant and should be removed. For example, below Phage therapy, Rhizome microbiome engineering, below Auxin, Jasmonic acid etc. Also, missing literature citing should be added into these table references.
RESPONSE: We apologize for any confusion. We corrected/removed these lines in Tables 2 and 4 and cited/added new references.
Thank you again for reviewing our manuscript favorably, and eagerly awaiting your positive response.
Best regards,
Sincerely,
Tika Adhikari

Reviewer 2 Report (New Reviewer)
Comments and Suggestions for Authors
Manuscript plants-2816729 describes the effects of rhizosphere microbiomes on the development of tomato plant defenses against major soil-borne diseases. This area of omics technology is very promising in agrobiotechnology for the development of sustainable agriculture and reducing crop losses due to soil-borne diseases. Understanding the functioning of plant microbiomes and the mechanisms of plant defense against pathogens can open up great prospects for increasing the productivity of crops. The presented manuscript, despite the relevance of the topic, requires major revision to improve the quality of the text.
Major remarks:
1. The authors describe soil-borne diseases of tomato well, but the description of rhizosphere microbiomes and their functioning in the manuscript is fragmentary, with not always adequate references (see below).
2. I have the impression that the authors give different meanings to the term “microbiome” in different parts of the manuscript. In some places, this is a stable community of microbes specific to a plant species (or even variety); in other places, inoculation of plants with pure or mixed cultures alters the microbiome. It would be useful in the Introduction section to define the terms “microbiome” and “microbiome-mediated strategies” and to consider the references in these terms.
3. Instead of (or in addition to) the Conclusion section, it would be useful to supplement the manuscript with a section “Prospects for the use of microbiome-mediated strategies”, in which the authors’ opinion should be given on: i) what the use of microbiome-mediated strategies should be; ii) what a barrier to such application at this stage of modern technologies; and iii) what future research could contribute to the successful and more effective use of microbiome-mediated strategies in agribiotechnology.
Minor remarks:
1) The “Introduction” section is very short and consists of sentences with little connection to each other. In addition, I have comments on almost every sentence in this section:
i) “Tomato is an essential vegetable produced globally, second only to potatoes.” According to the FAO, in 2022 (the latest year for which statistics are available), the tomato ranked third among vegetables after potato and cassava.
ii) “Asia is the largest producer of tomatoes, accounting for 62% of the world's production, followed by North America (13.2%), Europe (12.6%), and Africa (12%) [1]. The top four tomato producers are China (mainland) with 62.8 million tons, India with 19 million tons, Turkey with 12.8 million tons, and the USA with 10.9 million tons [1].” These data are taken almost unchanged from the article by Sotelo-Cardona et al., 2021 [ref. 1]. But the authors of the article [1] refer to the FAO website. The data given corresponds to 2019. Now there is information for 2022, which differs from the above. For a paper scheduled for publication in 2024, it would be helpful to provide updated information.
iii) “The genetic diversity of wild relatives can be harnessed to develop tomato varieties with desired traits [2].” This sentence is not connected with either previous or subsequent sentences. In addition, the link to the dissertation [ref. 2] seems doubtful to me. iv) “However, tomatoes are vulnerable to various environmental factors, including drought, salinity, extreme temperatures, pests, and diseases, and these threats can harm sustainability and food security [3].” – reference [3] is associated only with the last part of the sentence, and for “drought, salinity, extreme temperatures, pests” no reference is provided. v) “Tomato crop productivity and nutritional quality can be improved with microbiome-mediated strategies [4,5].” These references do not confirm improvement in tomato productivity with microbiome-mediated strategies.
2) Lines 45–46: “One of the most significant challenges farmers face is the prevalence of soil-borne diseases...” In the Introduction section (lines 38–39), various negative factors for tomato are given. However, here, without explanation, comparison or any reference, the authors claim that the most important negative factor is soil-borne diseases.
3) Lines 65–66: “We must improve our understanding of how pathogens spread, the infection process” - this is true, but this aspect does not find any further continuation in the text of the manuscript.
4) Table 2 should be given after the first link to it (line 92).
5) Line 128: Bacteria of the genera Bacillus and Paenibacillus belong to the phylum Firmicutes; therefore, the list “Bacillus, Firmicutes, Flavobacterium, Paenibacillus” is not correct.
6) Line 195: “KRN cell walls”? Does this mean “RKN cell walls”? Does this mean that nematode [animal] cells have a cell wall?
7) Section 4 is titled “The Direct and Indirect Beneficial Effects of Microbiome on Tomato”. However, it does not provide information on direct or indirect effects. Figure 2 lists direct and indirect effects on major diseases. However, it is not clear on what basis the authors divided the effects into direct and indirect. For example, antagonism is classified as having direct effects, and the production of antibiotics is classified as indirect. Why? Why is stimulation of plant growth classified as a direct effect against major soil-borne diseases? The references indicated by the authors [8, 87] do not make Figure 2 clearer. The authors should explain on the basis of whether they consider this or that effect to be direct or indirect.
8) Give the definition of the abbreviation VOC the first time it is mentioned in the text.
9) Section 8.3 is titled “Biochemical Approach”, but the influence of microbes on this aspect is given in one sentence with one reference (line 376-377 - “It is well known that plant growth-promoting microbiomes (PGPM) can influence the soil's oxidative, pH, and ion levels [146].) This one requires some explanation. If the authors use “It is well known,” then it can be assumed that there are many sources discussing the influence of microbiomes on soil chemistry. Describe here, if known, the mechanisms of such influence.
10) Section 8.4: Check the correctness of the literature references. For example, in the text on lines 389–393, about salt stress, reference 151 is given. Article by Roy et al., 2019 [ref. 151] does not discuss salt stress but describes an experiment that is accompanied by references 152 and 153 in this article.
11) The text of Section 8.6 requires changes. Show how Culturomics can be useful for managing major soil-borne diseases of tomato.
12) Figures 1C and 1D were not made by the authors of the manuscript. Nowhere is it stated that the figures are reproduced with the permission of their authors (For example, in the Conflicts of Interest section).
Author Response
January 17, 2024
Mr. Lance Cheng
MDPI Plants Editorial Office
St. Alban-Anlage 66, 4052 Basel, Switzerland
Dear Mr. Cheng:
Greetings from NC State University.
We are pleased to resubmit the manuscript "Microbiome-mediated Strategies to Manage Major Soil-borne Diseases of Tomato" with the Manuscript ID: plants-2816729 for publication. We appreciate your efforts and time in reviewing our manuscript and providing valuable feedback that helped us improve it. We have considered the comments from you and the other two reviewers and made several positive changes in this revised version. Some of the main changes include:
- Restructuring the Abstract, and Introduction.
- Polish and clarify the language in all sections to avoid any confusion.
- We added new Fig. 1C and 1D and revised Fig. 2 and Fig. 3 to better reflect the microbiome's roles in tomato diseases and plant immunity.
- Describe the direct and indirect effects of microbiomes in mitigating major soil-borne diseases of tomatoes. Additionally, we have included several new references in the revised manuscript.
We edited the manuscript before submission and attached the following documents.
- A rebuttal letter that responds to each point raised by the two reviewers.
- A "revised manuscript” - changes are highlighted in yellow.
I appreciate your consideration. We look forward to hearing back from you soon.
Best regards,
Sincerely
Tika Adhikari
******************************************************************************
Responses to Reviewer #2’s Comments:
Comments and Suggestions for Authors:
COMMENTS: Manuscript plants-2816729 describes the effects of rhizosphere microbiomes on the development of tomato plant defenses against major soil-borne diseases. This area of omics technology is very promising in agrobiotechnology for the development of sustainable agriculture and reducing crop losses due to soil-borne diseases. Understanding the functioning of plant microbiomes and the mechanisms of plant defense against pathogens can open up great prospects for increasing the productivity of crops. The presented manuscript, despite the relevance of the topic, requires major revision to improve the quality of the text.
RESPONSE: Thank you for your excellent comments, valuable feedback and suggestions.
COMMENT: The authors describe soil-borne diseases of tomato well, but the description of rhizosphere microbiomes and their functioning in the manuscript is fragmentary, with not always adequate references (see below).
RESPONSE: Thank you for pointing out this critical point. The introduction, other sections, and Figures have been revised accordingly, and references have been added.
COMMENT: I have the impression that the authors give different meanings to the term “microbiome” in different parts of the manuscript. In some places, this is a stable community of microbes specific to a plant species (or even variety); in other places, inoculation of plants with pure or mixed cultures alters the microbiome. It would be useful in the Introduction section to define the terms “microbiome” and “microbiome-mediated strategies” and to consider the references in these terms.
RESPONSE: We defined "microbiome" and "microbiome-mediated strategies" in the Introduction section and cited appropriate references. Your feedback was very constructive and we appreciate your insightful comments.
COMMENT: Instead of (or in addition to) the Conclusion section, it would be useful to supplement the manuscript with a section “Prospects for the use of microbiome-mediated strategies,” in which the authors’ opinion should be given on: i) what the use of microbiome-mediated strategies should be; ii) what a barrier to such application at this stage of modern technologies; and iii) what future research could contribute to the successful and more effective use of microbiome-mediated strategies in agribiotechnology.
RESPONSE: Thank you for raising this critical point. We respect your suggestion. We combined the Conclusion section with your suggested subheading. The new section is “Prospects for the use of microbiome-mediated strategies.” Kindly see below:
Microbiome-based strategies have the potential to manage soil-borne pathogens in tomato agri-biotechnology effectively. The main approach should be to explore the specificity of microbial strains and their effects on soil-borne pathogens across diverse environments to create robust biological-based disease management solutions. However, more research is needed to discover the untapped potential of soil microbiomes, their biological functions, stability, scalability, and how they work together as a community. These approaches help improve tomato disease resistance and nutrient uptake while reducing chemical dependence.
Technological barriers hinder the manipulation of microbial communities, and there is a limited understanding of microbial interactions. Future research should prioritize unraveling plant-microbe interactions using advanced metagenomic, metabolomic, metatranscriptomic, and proteomic approaches and employing high-throughput sequencing technologies. We can use natural or engineered microbiomes or consortia to improve crop performance and mitigate the negative impacts of tomato soil-borne diseases. Manipulating root exudates and utilizing SynComs, which is the inherent potential of the native plant microbiome, can also be helpful. Combining microbiome engineering and plants with the best management practices is necessary to address climate change and growing populations, promote ecosystem sustainability, and improve food security.
- COMMENT: The “Introduction” section is very short and consists of sentences with little connection to each other. In addition, I have comments on almost every sentence in this section:
- i) “Tomato is an essential vegetable produced globally, second only to potatoes.” According to the FAO, in 2022 (the latest year for which statistics are available), the tomato ranked third among vegetables after potato and cassava.
RESPONSE: We found it quite helpful as we approached our revision. Now, we have restructured the Introduction precisely. Regarding tomato cultivation, it is second after potatoes is correct. We also checked some data and references. For instance, Ma, M.; Taylor, P.W.J.; Chen, D.; Vaghefi, N.; He, J.-Z. Major Soilborne Pathogens of Field Processing Tomatoes and Management Strategies. Microorganisms 2023, 11, 263.
- ii) COMMENT: Asia is the largest producer of tomatoes, accounting for 62% of the world's production, followed by North America (13.2%), Europe (12.6%), and Africa (12%) [1]. The top four tomato producers are China (mainland) with 62.8 million tons, India with 19 million tons, Turkey with 12.8 million tons, and the USA with 10.9 million tons [1].” These data are taken almost unchanged from the article by Sotelo-Cardona et al., 2021 [ref. 1]. But the authors of the article [1] refer to the FAO website. The data given corresponds to 2019. Now there is information for 2022, which differs from the above. For a paper scheduled for publication in 2024, it would be helpful to provide updated information.
RESPONSE: Thank you for bringing these issues to our attention. We appreciate your suggestion and have revised the text accordingly. We deleted several extraneous and unnecessary sentences from the previous manuscript and added new reference FAO website 2022.
iii) COMMENT: The genetic diversity of wild relatives can be harnessed to develop tomato varieties with desired traits [2].” This sentence is not connected with either previous or subsequent sentences. In addition, the link to the dissertation [ref. 2] seems doubtful to me. iv) “However, tomatoes are vulnerable to various environmental factors, including drought, salinity, extreme temperatures, pests, and diseases, and these threats can harm sustainability and food security [3].” – reference [3] is associated only with the last part of the sentence, and for “drought, salinity, extreme temperatures, pests” no reference is provided.
RESPONSE: We apologize for the inconvenience. This section had to be removed because there was insufficient evidence regarding tomato diseases.
- v) COMMENT: Tomato crop productivity and nutritional quality can be improved with microbiome-mediated strategies [4,5].” These references do not confirm improvement in tomato productivity with microbiome-mediated strategies.
RESPONSE: Thank you for bringing up this important point. We have now provided more detail in the revised manuscript below:
Tomato is attacked by over 200 different diseases caused by various pathogens and pests [Singh et al. 2017]. These threats can harm tomato sustainability and food security [3]. Historically, several management strategies have been employed to manage tomato soil-borne diseases. These include the use of resistant cultivars, chemicals (e.g., fungicides and soil fumigants), physical methods such as soil solarisation and soil heating, and cultural methods (e.g., crop rotation field sanitation) and biological control [Bawa 2016; Baysal-Gurel et al., Spadaro and Gullino 2005]. Among these methods, biological control has recently gained more attention due to its high efficiency, target-specificity, sustainability, and public acceptance (Singh et al., 2018; Sood et al., 2020].
- COMMENT: Lines 45–46: “One of the most significant challenges farmers face is the prevalence of soil-borne diseases...” In the Introduction section (lines 38–39), various negative factors for tomato are given. However, here, without explanation, comparison or any reference, the authors claim that the most important negative factor is soil-borne diseases.
RESPONSE: We apologize for any confusion caused and have removed the sentences. Please see below for the updated information.
Soil-borne diseases pose a critical threat to tomato production systems due to their persistent nature and impact on root systems [Cheng et al. 2021]. Among these, Fusarium wilt (FW), Verticillium wilt (VW), and root-knot nematodes (RKN) pathogens can survive in the soil for extended periods, hindering effective crop rotation [Panth et al. 2020]. Consequently, the diseases compromise root function, impeding water and nutrient absorption, leading to stunted growth and reduced yield
- COMMENT: Lines 65–66: “We must improve our understanding of how pathogens spread, the infection process” - this is true, but this aspect does not find any further continuation in the text of the manuscript.
RESPONSE: Thank you for bringing up this important point. We deleted these irrelevant sentences.
- COMMENT: Table 2 should be given after the first link to it (line 92).
RESPONSE: Corrected.
- COMMENT: Line 128: Bacteria of the genera Bacillus and Paenibacillus belong to the phylum Firmicutes; therefore, the list “Bacillus, Firmicutes, Flavobacterium, Paenibacillus” is not correct.
RESPONSE: We found it quite helpful and restructured these statements like this: The TRM1 strain of Flavobacteria's assembled genome was linked to carbohydrate metabolism [36]. Specific monosaccharide transporters indicated that TRM1 can lower the availability of sugars that R. solanacearum lectin binds to, thus reducing BW disease pressure [36, 37]. Another strategy to reduce the spread of BW disease in tomatoes caused by R. solanacearum and promote beneficial microbiomes was recommended to increase the soil pH to 6.45. [38]. This process can increase bacterial genera such as Bacillus and Paenibacillus belonging to the phylum Firmicutes and other beneficial genera like Flavobacteriium and Pseudomonas spp. [38, Brady et al. 2018, Zboralski et al. 2022].
- COMMENT: Line 195: “KRN cell walls”? Does this mean “RKN cell walls”? Does this mean that nematode [animal] cells have a cell wall?
RESPONSE: Per your suggestion, we have revised it like this: Additionally, these fungi produce hydrolytic enzymes such as chitinases and are responsible for the enzymatic degradation of chitin, a structural component of RKN [Gortari and Hours 2008].
- COMMENT: Section 4 is titled “The Direct and Indirect Beneficial Effects of Microbiome on Tomato”. However, it does not provide information on direct or indirect effects. Figure 2 lists direct and indirect effects on major diseases. However, it is not clear on what basis the authors divided the effects into direct and indirect. For example, antagonism is classified as having direct effects, and the production of antibiotics is classified as indirect. Why? Why is stimulation of plant growth classified as a direct effect against major soil-borne diseases? The references indicated by the authors [8, 87] do not make Figure 2 clearer. The authors should explain on the basis of whether they consider this or that effect to be direct or indirect.
RESPONSE: The revised manuscript has been improved by adding some examples of direct and indirect effects, and Figure 2 has been revised per your suggestion. Thank you!
Microbiomes can enhance plant growth through direct and indirect effects [Glick 2012, 2020]. Direct effects can occur when the microbiome produces hormones like auxin (IAA), cytokinin, and gibberellin, or solubilizes nutrients like phosphorus, iron, or potassium, which in turn results in increased plant biomass, yield, and improved nutritional value of fruits and seeds [Singh et al. 2017, Gamalero and Glick 2019]. Indirect effects of the microbiome help reduce the negative impact of soil-borne pathogens [Glick 2012]. This is achieved through ACC deaminase that lowers ethylene levels, production of pathogen-inhibiting antibiotics and bacteriocins, synthesis of pathogen fungal cell wall-degrading enzymes and plant defense genes (e.g. chitinases, glucanases), out-competing pathogens for nutrients (e.g. siderophore production) or space, synthesis of fungal pathogen-inhibiting hydrogen cyanide, and induction of systemic resistance mechanisms such ISR, and SAR [Glick 2012, 2020, Gamalero and Gkick 2019, Olanrewaju et al. 2017].
- COMMENT: Give the definition of the abbreviation VOC the first time it is mentioned in the text.
RESPONSE: Corrected as Volatile organic compounds (VOCs).
- COMMENT: Section 8.3 is titled “Biochemical Approach”, but the influence of microbes on this aspect is given in one sentence with one reference (line 376-377 - “It is well known that plant growth-promoting microbiomes (PGPM) can influence the soil's oxidative, pH, and ion levels [146].) This one requires some explanation. If the authors use “It is well known,” then it can be assumed that there are many sources discussing the influence of microbiomes on soil chemistry. Describe here, if known, the mechanisms of such influence.
RESPONSE: Thank you for the constructive suggestion. We have revised this “Biochemical Approach section” extensively. Kindly see below:
Plants release chemical signals and substances from their roots to enhance the root microbiome, promoting a more beneficial microbiome makeup [135]. These exudates comprise organic acids, amino acids, sugars, alcohols, phenolics, polypeptides, enzymes, proteins, and hormones [135]. Root exudates shape plant-environment interactions [Reinhold-Hurek et al., 2015]. Analyzing these exudates through advanced metabolomics technique is an advantageous tool for investigating the interplay between plant-soil-borne pathogen interactions and optimizing plant-soil dynamics [Oburger et al., 2022; Canarini et al., 2016; Casas and Matamoros et al., 2021].
Plants can selectively promote the growth of beneficial microbiomes by changing the profile of metabolites released from the roots. This method enhances nutrient absorption, disease resistance, and overall plant health by promoting a more beneficial microbiome makeup [145]. The soil microbiome, through nutrient cycling (Yadav et al. 2021), and pH regulation (Cheng et al. 2021), improves soil structure significantly, which influences soil chemistry and mitigates soil-borne diseases in tomatoes (Nuzzo et al., 2020). Microbial activities, including decomposition, mineralization, and chemical signaling, enhance plant health and contribute to a balanced and beneficial rhizosphere [Reinhold-Hurek, et al., 2015].
Plant growth-promoting microbiomes (PGPM) influence the soil's oxidative, pH, and ion levels, shaping the chemical environment conducive to plant growth and development [146]. These findings lay the groundwork for exploring microbiome-based strategies to optimize soil conditions and combat soil-borne pathogens. For instance, studies reveal that reducing soil pH compromises the natural defense mechanisms of the soil microbiome against pathogenic Fusarium infections [Li et al. 2023].
- COMMENT: Section 8.4: Check the correctness of the literature references. For example, in the text on lines 389–393, about salt stress, reference 151 is given. Article by Roy et al., 2019 [ref. 151] does not discuss salt stress but describes an experiment that is accompanied by references 152 and 153 in this article.
RESPONSE: Thank you for pointing out these references. Irrlevant references (abiotic or salt stresses) deleted.
- COMMENT: The text of Section 8.6 requires changes. Show how Culturomics can be useful for managing major soil-borne diseases of tomato.
RESPONSE: There are limited studies on soil-borne diseases of tomatoes. The following paragraph has been added:
Although culturomics has emerged as a novel approach combining next-generation sequencing and using a wide range of culture media [150, 158], it has not yet been extensively used to describe the tomato microbiome. One study used tomato extract to cultivate bacterial taxa Oenococcus oeni [Sarhan et al. 2019]. In another study, various vegetables, including tomatoes, were sampled and analyzed, and the rhizosphere soils of various vegetables, including tomatoes, were sampled and analyzed to understand microbial diversity [Lian et al. 2023].
- COMMENT: Figures 1C and 1D were not made by the authors of the manuscript. Nowhere is it stated that the figures are reproduced with the permission of their authors (For example, in the Conflicts of Interest section).
RESPONSE: Both Figures 1C and 1D have been replaced and secured contributors’ permission. An acknowledgment section was created and mentioned like this:
Acknowledgments: The authors thank Inga Meadows and Dr. Weinmin Ye for contributing pictures in Figure 1C and 1C, respectively, which were reproduced with their permission.
Thank you again for your consideration, and I look forward to your positive response.
Tika Adhikari

Round 2
Reviewer 2 Report (New Reviewer)
Comments and Suggestions for Authors
The revised manuscript is a significant improvement over the first version. The authors fully answered all my questions and comments. For the revised manuscript, I have only a few technical comments that can be used to proofread the article.
Minor remarks:
Lines 31-33 - “The top four tomato producers are China (mainland) with 67.5 million tons, India with 21.1 million tons, Turkey with 13 million tons, and the USA with 10.4 million tons” – FAO data (https://www.fao.org/faostat/en/#data/QCL) for the 2022 year: “The top four tomato producers are China (mainland) with 68.2 million tons, India with 20.7 million tons, Turkey with 13 million tons, and the USA with 10.2 million tons”.
Line 57 – “Root microbiota is derived...” would be more accurate to say, “Root microbiota is mostly derived...”.
Table 4, line “Auxin (IAA)” – correct would be “Actinobacterial Streptomycetaceae family and Bacillus spp.” But in general, “Actinobacterial Streptomycetaceae family” is overly detailed information for bacteria of the Streptomycetaceae family; just “Streptomycetaceae” would be enough.
Lines 523-524 – “One study used tomato extract to cultivate bacterial taxa Oenococcus oeni [209].” In this sentence, replace “tomato extract” with “tomato juice” and “bacterial taxa Oenococcus oeni” with “bacterium Oenococcus oeni”. Additionally, "One study used..." implies a reference to experimental work. Therefore, instead of linking to a review, it is better to provide a link to the original source (https://doi.org/10.1099/00221287-48-3-431).
Line 555 – Replace “Figure 1C and 1C, respectively” with “Figure 1C and 1D, respectively”.
Author Response
Review Report Form – Reviewer 2 (#2)
Comments and Suggestions for Authors
Comment: The revised manuscript is a significant improvement over the first version. The authors fully answered all my questions and comments. For the revised manuscript, I have only a few technical comments that can be used to proofread the article.
Response: Thank you for your insightful comment. We are grateful to you for your constructive suggestions – which help to improve this manuscript.
Minor remarks:
Comment: Lines 31-33 - “The top four tomato producers are China (mainland) with 67.5 million tons, India with 21.1 million tons, Turkey with 13 million tons, and the USA with 10.4 million tons” – FAO data (https://www.fao.org/faostat/en/#data/QCL) for the 2022 year: “The top four tomato producers are China (mainland) with 68.2 million tons, India with 20.7 million tons, Turkey with 13 million tons, and the USA with 10.2 million tons”.
Response: Corrected as per your suggestion. The top four tomato producers are China (mainland) with 68.2 million tons, India with 20.7 million tons, Turkey with 13 million tons, and the USA with 10.2 million tons, according to FAO data (https://www.fao.org/faostat/en/#data/QCL) for the 2022 year [1, 2].
Comment: Line 57 – “Root microbiota is derived...” would be more accurate to say, “Root microbiota is mostly derived...”.
Response: Thank you. We changed it to ‘Root microbiota is mostly derived...’.
Comment: Table 4, line “Auxin (IAA)” – correct would be “Actinobacterial Streptomycetaceae family and Bacillus spp.” But in general, “Actinobacterial Streptomycetaceae family” is overly detailed information for bacteria of the Streptomycetaceae family; just “Streptomycetaceae” would be enough.
Response: We corrected it to ‘Streptomycetaceae’.
Comment: Lines 523-524 – “One study used tomato extract to cultivate bacterial taxa Oenococcus oeni [209].” In this sentence, replace “tomato extract” with “tomato juice” and “bacterial taxa Oenococcus oeni” with “bacterium Oenococcus oeni”. Additionally, "One study used..." implies a reference to experimental work. Therefore, instead of linking to a review, it is better to provide a link to the original source (https://doi.org/10.1099/00221287-48-3-431).
Response: We corrected it as per your suggestion. Thank you. ‘One study used tomato juice to cultivate the bacterium Oenococcus oeni [209; https://doi.org/10.1099/00221287-48-3-431]’.
Comment: Line 555 – Replace “Figure 1C and 1C, respectively” with “Figure 1C and 1D, respectively”.
Response: Thank you for pointing out the typo. We corrected it. The authors thank Inga Meadows and Dr. Weinmin Ye for contributing pictures in Figures 1C and 1D, respectively, which were reproduced with their permission.

This manuscript is a resubmission of an earlier submission. The following is a list of the peer review reports and author responses from that submission.
Round 1
Reviewer 1 Report
Comments and Suggestions for Authors
This review lacks a specific focus on tomatoes and is composed of various paragraphs stitched together. Only a few examples emphasize tomatoes, while the rest provide more general information with limited tomato-centric focus. Another concern pertains to the abundance of information without proper references and an exaggeration of facts.
The title needs a revision to accurately reflect the provided information, its connection to food security, and the novel insights it offers.
Other topic specific comments:
When considering the importance of crops, it's vital to exercise caution. Exaggeration for the sake of publication should be avoided. The first section should be condensed, omitting unnecessary details about tomato types.
While the use of agrochemicals can indeed enhance production, it may also create issues related to resistance and environmental impact.
Table 1 requires improvement. It should contain relevant information about the role of microbes in specific diseases and how they are used.
In lines 63-64, specific tomato citations and examples should be included.
Section 5 on Artificial Intelligence (AI) to Measure Root Architecture should be linked to the role of microbes in root architecture.
Figure 1 currently conveys general physiology, but it should illustrate the mechanisms through which microbes contribute to plant immunity.
Line 210 is unclear and needs to be rephrased with pertinent information.
Regarding Figure 2, the figure title does not accurately reflect its content. Specify the potential benefits you are referring to in this figure.
Table 4 discusses metabolomics and other omics approaches, which should be clarified as genomic approaches as well.
Author Response
NOTE: "Please see the attachment."
November 01, 2023
Plants, MDPI AG, St. Alban-Anlage 66
Open Access, 4052 Basel, Switzerland
Dear Editor and Reviewer 1:
Greetings from NC State University.
We are pleased to submit the revised version of our manuscript titled "Microbiome-mediated Strategies to Manage Major Soil-borne Diseases of Tomato for Food Security" with the Manuscript ID: plants-2671375 for publication. We appreciate your efforts and time in reviewing our manuscript and providing valuable feedback that helped us improve it. We have considered the comments from you and the other two reviewers and made several positive changes in this revised version. Some of the main changes include:
- Restructuring the Title, Abstract, and Introduction.
- Polishing and clarifying the language in all sections to avoid any confusion.
- Revising Fig. 1 and Fig. 2 better to reflect the roles of microbiome contributions to plant immunity.
- Restructuring the title and content of Fig. 2 to reflect the potential benefits accurately.
- Highlighting the challenges that significant diseases pose to tomato production in Table 1 and delving into the topic of the roles of microbiomes in mitigating the effects of four major soil-borne diseases of tomatoes, namely bacterial wilt, Verticillium wilt, Fusarium wilt, and Root-knot nematode under subheading 3. We provide insights into how microbiomes can help in reducing the impact of these diseases. Additionally, we have included several new references in the revised manuscript.
We have also edited the manuscript before submission and attached the following documents.
- A rebuttal letter that responds to each point raised by the two reviewers.
- A marked-up copy of the manuscript using highlighting yellow-colored text, which is labeled as "Revised Manuscript with Track Changes."
I appreciate your consideration. We look forward to hearing back from you soon.
Best regards,
Sincerely
Tika Adhikari
***********************************************************************************
Responses to Reviewer #1’s Comments:
Review Report Form 1
Comments and Suggestions for Authors:
GENERAL COMMENTS: This review lacks a specific focus on tomatoes and comprises various paragraphs stitched together. Only a few examples emphasize tomatoes, while the rest provide more general information with limited tomato-centric focus. Another concern pertains to the abundance of information without proper references and an exaggeration of facts.
RESPONSE: We apologize for any confusion caused and have revised the statement with new references.
COMMENT: The title needs a revision to accurately reflect the provided information, its connection to food security, and the novel insights it offers.
RESPONSE: Thank you for your insightful comments and excellent ideas. As per your suggestion, we have revised the manuscript's title. The new title is "Microbiome-mediated Strategies to Manage Major Soil-borne Diseases of Tomato for Food Security.”
"Other topic specific comments:
COMMENT: When considering the importance of crops, it's vital to exercise caution. Exaggeration for the sake of publication should be avoided. The first section should be condensed, omitting unnecessary details about tomato types.
RESPONSE: Thank you for bringing these issues to our attention. We appreciate your suggestion and have revised the text accordingly. In this revision, we deleted several extraneous and unnecessary sentences from the introduction.
COMMENT: While the use of agrochemicals can indeed enhance production, it may also create issues related to resistance and environmental impact.
RESPONSE: We found it quite helpful as we approached our revision. Now, we included in Abstract Page, Lines 11-13: …. “However, tomato production encounters several challenges, particularly concerning the use of pesticides to control pests and diseases. The application of agrochemicals raises concerns regarding chemical residues, pesticide resistance, and negative environmental impacts raise concerns about chemical residues, pesticide resistance, and negative environmental impacts.”
COMMENT: Table 1 requires improvement. It should contain relevant information about the role of microbes in specific diseases and how they are used.
RESPONSE: Thank you for your valuable feedback and suggestions. In Table 1, we have outlined the challenges that major diseases pose to tomato production. Additionally, in subheading 3, we have delved into the topic of the roles of microbiomes in mitigating four major soil-borne diseases of tomatoes, namely Bacterial wilt, Verticillium wilt, Fusarium wilt, and Root-knot nematode. Our manuscript provides insights into how microbiomes can be effective in reducing the impact of these diseases. We have also included several new references in the revised version.
COMMENT: In lines 63-64, specific tomato citations and examples should be included.
RESPONSE: The revised manuscript has been improved by removing unclear sentences based on the great suggestion you provided. Thank you!
COMMENT: Section 5 on Artificial Intelligence (AI) to Measure Root Architecture should be linked to the role of microbes in root architecture.
RESPONSE: We apologize for any confusion caused. We revised subheadings 4 and 5 and removed irrelevant sentences from the manuscript.
COMMENT: Figure 1 currently conveys general physiology, but it should illustrate the mechanisms through which microbes contribute to plant immunity. New Fig 1 is below:
RESPONSE: We are sorry for any confusion caused. We have removed these sentences in the updated manuscript.
COMMENT: Line 210 is unclear and needs to be rephrased with pertinent information.
RESPONSE: Thank you for bringing up this important point. We have now provided more detail in the revised manuscript.
COMMENT: Regarding Figure 2, the figure title does not accurately reflect its content. Specify the potential benefits you are referring to in this figure.
RESPONSE: We restructured the title and content of Fig. 2 to reflect the potential benefits accurately. The new title of Fig. 2 is below:
Figure 2. An integrated plant microbiome research to understand the beneficial effects of the plant-associated microbiome. Microbiome engineering presents exciting new opportunities for various application methods and multi-omics technologies and can offer multiple benefits, including plant growth promotion, health, and immunity against pathogens. The benefits provided by the microbiome can be initiated in any part of the plant and transmitted to other parts of the plant through plant-mediated transport or signals, thereby triggering plant immunity.
COMMENT: Table 4 discusses metabolomics and other omics approaches, which should be clarified as genomic approaches as well.
RESPONSE: We revised the title to include Omics Approaches, amplicon sequencing, and metagenomics, hoping to meet the requirements. I hope the revised manuscript has clarified all the necessary points for the reviewers to take the next step. I also greatly appreciate your assistance in improving the readability of our submission. Thank you again for considering our work, and I am eagerly awaiting your positive response.
Best regards,
Sincerely,
_________________________________
Tika Adhikari, Ph. D.
Principal Research Scholar & Adjunct Professor
Department of Entomology and Plant Pathology
North Carolina State University
Raleigh, NC 27695

Reviewer 2 Report
Comments and Suggestions for Authors
This is potentially a very useful manuscript. However, in its present form it suffers form too many statements that have little information to the reader other than if they wanted information they should read the citations mentioned by the authors.
For instance in the section Impact of Tomato Pathogens on Food Security and the Importance of Microbiome to Increase Food Security; there are many generalities of the role of the microbiome but the only one concrete example provided is the possible role of Trichoderma interacting with and insect.
The manuscript is full of statements that says tot he reader there is a lot good information available and we found it and now you find it.
Examples are Line 183: There are feedback loops as certain microbes trigger the production of specific metabolites, which further influence microbial colonization [46]. What are these feed back loops?
Additionally, the composition of root exudates, influenced by environmental factors and microbial presence, plays a dynamic role in shaping the root microbiome. What are these root exudates?
It is essential to understand these intricate interactions to optimize agricultural practices and enhance plant-microbe partnerships [47]. Tell me how I should go about understanding these interactions.
Line 194: The "cry-for-help" hypothesis suggests that plants alter root exudates to signal the microbial community when exposed to pathogens, encouraging the growth of beneficial microbes [49]. Great stuff - what are the signals and how do they impact the community? Should I read ref 49 or should you tell me?
The manuscript is full of such statements , which are interesting but remain unanswered.
I suggest rewriting the manuscript to focus on areas that have shown concreted evidence that managing the microbiome either by the plant, the environment, or by external means has impacted in a meaningful way our abilities to produce a health tomato crop.
Author Response
NOTE: "Please see the attachment."
November 01, 2023
Plants, MDPI AG, St. Alban-Anlage 66
Open Access, 4052 Basel, Switzerland
Dear Editor and Reviewer 2:
Greetings from NC State University.
We are pleased to submit the revised version of our manuscript titled "Microbiome-mediated Strategies to Manage Major Soil-borne Diseases of Tomato for Food Security" with the Manuscript ID: plants-2671375 for publication. We appreciate your efforts and time in reviewing our manuscript and providing valuable feedback that helped us improve it. We have considered the comments from you and the other two reviewers and made several positive changes in this revised version. Some of the main changes include:
- Restructuring the Title, Abstract, and Introduction.
- Polishing and clarifying the language in all sections to avoid any confusion.
- Revising Fig. 1 and Fig. 2 better to reflect the roles of microbiome contributions to plant immunity.
- Restructuring the title and content of Fig. 2 to reflect the potential benefits accurately.
- Highlighting the challenges that significant diseases pose to tomato production in Table 1 and delving into the topic of the roles of microbiomes in mitigating the effects of four major soil-borne diseases of tomatoes, namely bacterial wilt, Verticillium wilt, Fusarium wilt, and Root-knot nematode under subheading 3. We provide insights into how microbiomes can help in reducing the impact of these diseases. Additionally, we have included several new references in the revised manuscript.
We have also edited the manuscript before submission and attached the following documents.
- A rebuttal letter that responds to each point raised by the two reviewers.
- A marked-up copy of the manuscript using highlighting yellow-colored text, which is labeled as "Revised Manuscript with Track Changes."
I appreciate your consideration. We look forward to hearing back from you soon.
Best regards,
Sincerely
Tika Adhikari
************************************************************************************
Responses to Reviewer #2’s Comments
Review Report Form 2
Comments and Suggestions for Authors:
GENERAL COMMENTS: This is potentially a very useful manuscript. However, in its present form it suffers form too many statements that have little information to the reader other than if they wanted information they should read the citations mentioned by the authors.
RESPONSE: We apologize for any confusion caused by our original statement and appreciate you taking the time to review our manuscript. Thank you for your insightful comments and agree with your assessment. We found your feedback constructive and clarified it in the revised manuscript.
COMMENTS: For instance in the section Impact of Tomato Pathogens on Food Security and the Importance of Microbiome to Increase Food Security; there are many generalities of the role of the microbiome but the only one concrete example provided is the possible role of Trichoderma interacting with and insect.
RESPONSE: Thank you for bringing up these important points. We agree with you. In the revised manuscript, we have partitioned these points into subheadings 2 and 3, and we have included major diseases of tomatoes in Table 1. For example, subheading 2 briefly describes "Major Tomato Diseases: Threats to Food Security," Table 1 contains "Major Diseases Pose Challenges to Tomato Production," and subheading 3 highlights, "Insight into the Roles of Microbiomes to Mitigate Major Soil-Borne Diseases of Tomatoes." Please note that our review paper mainly focuses on key soil-borne diseases only.
COMMENT: The manuscript is full of statements that says to the reader there is a lot good information available and we found it and now you find it.
RESPONSE: Thank you for bringing these issues to our attention. We appreciate your suggestion and have revised the text. We deleted unnecessary sentences from the text and cited several examples related to microbiomes.
COMMENT: Examples are Line 183: There are feedback loops as certain microbes trigger the production of specific metabolites, which further influence microbial colonization [46]. What are these feed back loops?
RESPONSE: Thank you for bringing up this critical point. We appreciate your suggestion, and as a result, we have removed the statement from the discussion.
COMMENT: Additionally, the composition of root exudates, influenced by environmental factors and microbial presence, plays a dynamic role in shaping the root microbiome. What are these root exudates?
RESPONSE: We have revised the following paragraph for clarity and made corrections to any spelling, grammar, and punctuation errors. New references have been highlighted with the color yellow:
“Plants release compounds, such as phenolics, amino acids, nucleotides, sugars, terpenoids, polysaccharides, and proteins, which are low and high molecular weight [46]. Although the composition and quantity of these compounds differ depending on the plant species, growth stage, and presence of stress factors, they contribute to nutrient availability by promoting microbial growth and solubilizing minerals through root exudates [46]. Additionally, these compounds are essential in plant defense mechanisms that help to inhibit pathogens and promote resistance (Pang et al. 2021). Plant roots also secrete secondary metabolites (SMs) into the soil, which affect the proliferation or suppression of specific microbiomes around the host plant's root (Korenblum et al. 2020). Recent studies have shown that particular SMs, including coumarin, triterpenes, flavonoids, benzoxazinoids, and phytohormones, can shape the composition and functions of the plant's microbiome (Pang et al. 2021).
The exudation of metabolites is also linked to microbiome functions and can shape microbiome composition [Pang et al. 2021; 47; 48]. Some prominent metabolites include glucosinolates, flavonoids, coumarins, benzoxazinoids, triterpenes, and defense-related phytohormones jasmonate and salicylate [49]. Plants can adapt to pathogen infection by altering the chemistry of their root exudation [49], which may help them to assemble microbiomes that promote plant health (Trivedi et al. 2020).”
COMMENT: It is essential to understand these intricate interactions to optimize agricultural practices and enhance plant-microbe partnerships [47]. Tell me how I should go about understanding these interactions.
RESPONSE: Thank you for pointing out this issue. We have carefully reviewed your concern and tried to address it in reference number 47. Kindly refer to our above response for further information.
COMMENT: Line 194: The "cry-for-help" hypothesis suggests that plants alter root exudates to signal the microbial community when exposed to pathogens, encouraging the growth of beneficial microbes [49]. Great stuff - what are the signals and how do they impact the community? Should I read ref 49 or should you tell me?
RESPONSE: We apologize for any confusion caused. We revised this paragraph entirely and new references have been highlighted with the color yellow:
“Plants rely on beneficial soil microbiomes to help them deal with various stresses (Rolfe et al. 2019). Therefore, they must recruit, activate, and assemble microbiomes to protect and enhance their growth (Liu et al. 2021). In high-intensity production systems that rely on successive crop monocultures, disease-suppressive soil activity can develop (Kwak & Weller 2013). For instance, the take-all decline occurs when wheat is grown continuously in soil infested with the pathogenic fungus Gaeumannomyces graminis var. tritici (Kwak & Weller 2013). Initially, this leads to an increase in disease, but over time, there is a progressive decline in disease. Host plants can actively recruit or stimulate biocontrol agents to resist pathogen invasion, known as the 'cry for help' hypothesis (Liu et al. 2021; Rolfe et al. 2019). Studies have confirmed that enriching soil microbiomes can suppress diseases (Liu et al 2021). Additionally, plants differ in physiology and immune responses to pathogen invasions (Liu et al. 2020). The molecular mechanisms and chemical basis of the ‘cry for help’ strategy may vary depending on the plant species [50]. Disease-resistant plant genotypes also accumulate beneficial rhizosphere microbiomes, suggesting a link between immunity and microbiome composition (Liu et al. 2020). The interplay involves the subversion of plant hormone signaling and immune responses by specific microbial effectors [3; 51]. Furthermore, plant-derived vital hormones are released into the rhizosphere and can directly affect plant-interacting microbiomes and the root-associated microbiome (Table 3).”
COMMENT: The manuscript is full of such statements, which are interesting but remain unanswered.
RESPONSE: We deeply apologize for the previous statements. However, we have made the necessary corrections and improvements in the revised manuscript.
COMMENT: I suggest rewriting the manuscript to focus on areas that have shown concreted evidence that managing the microbiome either by the plant, the environment, or by external means has impacted in a meaningful way our abilities to produce a health tomato crop.
RESPONSE: We hope that the revised version of our manuscript has addressed all the concerns raised by you and the other reviewers. We greatly appreciate your valuable feedback which has helped in improving the readability of our submission. Thank you for your time and effort in reviewing our work. We are eagerly looking forward to your positive response.
Best regards,
Sincerely,
Tika Adhikari
_________________________________
Tika Adhikari, Ph. D.
Principal Research Scholar & Adjunct Professor
Department of Entomology and Plant Pathology
North Carolina State University
Raleigh, NC 27695

Round 2
Reviewer 1 Report
Comments and Suggestions for Authors
As per earlier suggested comments
Reviewer 2 Report
Comments and Suggestions for Authors
The manuscript has been much improved with very good identification of the concepts and novelty of the use and understanding of the plants microbiome to modulate plant health.
The section describing the impacts of the various pathogens is very poor and needs to be improved. I have made some suggestions in the word document uploaded.
Although grafting does increase the cost of plants significantly the vast majority of greenhouses use grafted plants and they are now produced probably by the hundreds of millions.
Comments on the Quality of English LanguageThe manuscript is well written and organized. I suggested changes to the section on the impacts of the major pathogens.